# Smartphone-delivered self-management for first-episode psychosis: the ARIES feasibility randomised controlled trial

Thomas Steare [iD],[1] Puffin O'Hanlon,[1] Michelle Eskinazi,[1] David Osborn,[1,2] Brynmor Lloyd-Evans,[1,2] Rebecca Jones,[1] Helen Rostill,[3,4] Sarah Amani,[5] Sonia Johnson [iD] [1,2]

[1]Division of Psychiatry, University College London, London, UK
[2]R&D Department, Camden and Islington NHS Foundation Trust, London, UK
[3]University of Surrey, Guildford, UK
[4]Surrey and Borders Partnership NHS Foundation Trust, Leatherhead, UK
[5]Early Intervention in Psychosis Programme (South of England), Oxford, UK

**Correspondence to**
Prof Sonia Johnson;
s.johnson@ucl.ac.uk

## ABSTRACT

**Objectives** To test the feasibility and acceptability of a randomised controlled trial (RCT) to evaluate a Smartphone-based self-management tool in Early Intervention in Psychosis (EIP) services.

**Design** A two-arm unblinded feasibility RCT.

**Setting** Six NHS EIP services in England.

**Participants** Adults using EIP services who own an Android Smartphone. Participants were recruited until the recruitment target was met (n=40).

**Interventions** Participants were randomised with a 1:1 allocation to one of two conditions: (1) treatment as usual from EIP services (TAU) or (2) TAU plus access to My Journey 3 on their own Smartphone. My Journey 3 features a range of self-management components including access to digital recovery and relapse prevention plans, medication tracking and symptom monitoring. My Journey 3 use was at the users' discretion and was supported by EIP service clinicians. Participants had access for a median of 38.1 weeks.

**Primary and secondary outcome measures** Feasibility outcomes included recruitment, follow-up rates and intervention engagement. Participant data on mental health outcomes were collected from clinical records and from research assessments at baseline, 4 months and 12 months.

**Results** 83% and 75% of participants were retained in the trial at the 4-month and 12-month assessments. All treatment group participants had access to My Journey 3 during the trial, but technical difficulties caused delays in ensuring timely access to the intervention. The median number of My Journey 3 uses was 16.5 (IQR 8.5 to 23) and median total minutes spent using My Journey 3 was 26.8 (IQR 18.3 to 57.3). No serious adverse events were reported.

**Conclusions** Recruitment and retention were feasible. Within a trial context, My Journey 3 could be successfully delivered to adults using EIP services, but with relatively low usage rates. Further evaluation of the intervention in a larger trial may be warranted, but should include attention to implementation.

**Trial registration** ISRCTN10004994.

## Strengths and limitations of this study

► Participant data were collected from a wide range of sources including questionnaires, patient records and from the app.
► Participants were followed up for a 12-month period; longer than the majority of feasibility trials investigating Smartphone apps for psychosis.
► We were not able to blind researchers or participants to their treatment allocation.
► The study recruited users of Early Intervention in Psychosis services that own an Android Smartphone, limiting sample representativeness.
► This is a feasibility study and therefore does not have the statistical power to conclude the effectiveness of the intervention.

## INTRODUCTION

Early Intervention in Psychosis (EIP) services have been established across the UK to provide care to adults during the 3 years following an initial episode of psychosis. There is evidence that such services are effective and cost-effective,[1 2] resulting in improvement in a range of outcomes, yet challenges remain. Relapse rates for EIP service users are high[3] particularly after discharge[4 5] and limited adherence with antipsychotic medication is common.[6] There are also difficulties accessing psychosocial interventions,[7] including supported self-management.

Illness self-management is an approach designed to support people to manage long-term health conditions by developing their ability to recognise and monitor symptoms and early warning signs of relapse, identify and avoid stressors, make plans for achieving their own recovery and effectively use coping strategies.[8] For people with psychosis, self-management tools have been shown to reduce psychological distress, improve medication adherence and reduce the likelihood of future hospital admissions.[9–11] In a recent meta-analysis, self-management interventions for severe mental illness were also found to have a significant benefit on patient-valued

outcomes of personal recovery, hope and self-efficacy.[12] Despite clinician-supported self-management programmes being mandated in current UK treatment guidelines for first-episode psychosis,[13] there is a lack of well-evaluated tools to support delivery within EIP services. There is a clear need to overcome implementation barriers affecting the delivery of self-management to those likely to benefit from it.[12] A potentially convenient and economical way of achieving this is via the use of digital technology such as Smartphones.[14]

Smartphones can run advanced software known as apps that hold promise as an effective tool to assist the monitoring and treatment of mental health problems. Smartphone ownership is rapidly growing worldwide[15] with a significant number of developed countries with ownership rates of more than 80%.[16] Adults with severe mental health problems have comparable Smartphone ownership rates with the general population,[17–19] and there is a growing consensus that adults with psychosis are open to using Smartphones to access mental health interventions.[20 21] Smartphones also provide high accessibility to the internet and are commonly carried on the person, meaning apps can be easily accessed at times and locations convenient for the user. Accordingly, Smartphones have the capacity to deliver time-unlimited mental health interventions, such as self-management tools, and ultimately the potential to increase access to effective care and reduce healthcare costs.[22] The benefits of Smartphone apps may also extend beyond the original treatment period with a community team and could be a valuable tool following discharge where the risk of relapse is increased.[4 5]

The majority of digital health interventions that have been developed for psychosis have been based on existing psychological therapies such as cognitive-behavioural therapy,[23 24] or other evidence-based interventions,[25 26] yet very little is known regarding their effectiveness when delivered in EIP services. A growing number of self-management apps for psychosis have been tested for feasibility and acceptability, including those delivered independently of a clinical setting and those embedded within clinical care.[27–29] These have shown promising levels of adoption and use in research contexts, yet little is known about their clinical efficacy.

To date, only one trial of a self-management app delivered in EIP services has published results regarding the intervention's impact on clinical outcomes.[30] In the proof-of-concept trial, an active self-management app 'Actissist' was found to confer benefits over a passive control app. The study suggests that participants who received Actissist had better outcomes regarding their mood and general and negative symptoms post-treatment in comparison with control participants. Actissist features a range of components including self-assessment questions focused on cognitive appraisals, emotions, behaviours and belief convictions and suggests appropriate coping strategies, but does not feature some major cornerstones of self-management such as relapse and recovery plans.

Regardless, results from this study suggest that such digital self-management interventions could potentially improve outcomes of people using EIP services. Further trials are needed before firm conclusions can be made regarding the feasibility of conducting randomised controlled trials (RCTs) in this field and of the therapeutic benefits of self-management apps for first-episode psychosis delivered in clinical settings.

We aimed to address this evidence gap by conducting a feasibility RCT of a supported self-management Smartphone app, 'My Journey 3', designed to help EIP service users recognise early warning signs of illness, recognise and monitor symptoms, and create plans for their recovery. My Journey 3 has been designed to be initially set up in EIP services and used with clinician support, but to also be suitable for independent use. The results of the feasibility RCT are a potential step towards a full-scale trial to assess the effectiveness of the intervention.

The objectives of this study were as follows:
1. To determine the acceptability of the My Journey 3 self-management app for use in an EIP service context.
2. To determine the feasibility of trial procedures for a definitive trial, including recruitment, intervention enrolment and trial attrition.
3. To test procedures for evaluating intervention engagement and participant outcomes.

## METHODS
### Design
The App to support Recovery In Early intervention Services (ARIES) study was an unblinded feasibility RCT with a nested qualitative study comparing a supported self-management Smartphone app (My Journey 3) in addition to treatment as usual (TAU), with a control group receiving TAU only. Participants were randomly allocated to one of the two trial arms in a 1:1 ratio. Since this was a feasibility trial, it was not designed to have sufficient statistical power to assess the effectiveness of the My Journey 3 intervention.

As the study was a feasibility trial, prospective registration was not required.[31] Further details of the methodology are available in the protocol paper.[32] We have followed the Consolidated Standards of Reporting Trials (CONSORT) statement extension for pilot and feasibility randomised trials for reporting.[33] A copy of the CONSORT checklist is provided as online additional file 1.

### Setting
The trial was conducted in six EIP services across three NHS Foundation Trusts in England. EIP services are multidisciplinary community mental health services that provide care coordination to people in the first 3 years of a first-episode psychosis, focusing on engagement, achieving social and clinical recovery and delivering a full range of pharmacological, psychological and social interventions.[34] The six EIP services as mandated in

England provide care for people up to the age of 65, with the potential for adults above the age range to access EIP services if clinically appropriate although these cases are rare. Two of the participating Trusts are located in inner London. The third Trust is located in a county outside of London with both urban and rural areas. Assessments were conducted face-to-face at EIP services, at participants' homes or at University College London.

## Participants

Participants were recruited from the participating EIP services over 7 months. We assumed a conservative 40% attrition rate and accordingly set the target sample size as 40 participants to ensure the trial retained 12 completer participants per group (as recommended to assess trial feasibility).[35] Participants were eligible if they were aged ≥16 years, had experienced at least one episode of psychosis, were currently on the caseload of an EIP service and owned a Smartphone with an Android operating system. People were excluded from the trial if they lacked capacity to consent to participation, were unable to communicate and understand English, or were considered by their EIP service to pose a high risk to researchers during meetings, even on NHS premises. Familiarity and competence in using digital technology or Smartphones was not an eligibility criterion.

## Recruitment strategy

Clinicians at the participating EIP services were briefed by the research team and were asked to make initial contact with eligible EIP service users. Clinicians explained the trial to service users and enquired whether the service user would be willing to speak to a researcher about participating in the trial. The researcher then made contact with eligible and potentially willing service users and arranged a face-to-face meeting where the trial was explained further. The researcher provided the trial information sheet (online additional file 2) and assessed the participant's capacity to provide informed consent. Service users had at least 24 hours after receiving the information sheet to consider their participation. Participants then gave written informed consent to take part, prior to completing the baseline assessment. No participants were recruited via online methods.

## Randomisation

Following the baseline assessment, participants were randomly allocated in a 1:1 ratio to either the intervention (n=20) or the control group (n=20) by an independent statistician. The treatment group had access to My Journey 3 in addition to TAU, while the control group received TAU only. An independent researcher held the allocation list and did not disclose participants' allocation to the trial researcher until after completion of the baseline assessments.

Due to the nature of the intervention, participants were not blinded to their group allocation. During the recruitment process, participants would have been aware that My Journey 3 was the intervention of interest. As a single researcher carried out the majority of data collection, it was not practical for the allocation of participants to be concealed from the research team. Participants were informed of their allocation by the researcher via a telephone call.

## Interventions
### My Journey 3

My Journey 3 is a Smartphone app developed for adults accessing EIP services. The aim of the intervention is to develop users' self-management skills to help them to achieve self-determined recovery goals and avoid future relapses. My Journey 3 is suitable for independent use, but also designed to be used with support from EIP service clinicians who will be able to assist with the completion of the self-management components and initial set-up. It is the developers' aspiration for My Journey 3 to be used initially in collaboration with EIP service clinicians, and for it to support continuing self-management after users have been discharged from EIP services.

The development of My Journey 3 has been through several iterations. The first version (My Journey 1) was created by Surrey and Borders Partnership NHS Foundation Trust with leadership from Sarah Amani, for EIP service users to track their symptoms, set reminders for appointments and share their progress with EIP service clinicians. In developing the current version of My Journey 3, we have drawn on existing paper-and-pen self-management intervention components[36 37] to allow users to track recovery goals and personalise relapse prevention plans—important cornerstones of illness self-management. The design and the content of My Journey 3 was led by a collaboration of researchers, digital health experts, EIP service clinicians and service users. A private app development company based in the UK (MyOxygen; https://myoxygen.uk) led the technical development of My Journey 3. To limit costs, My Journey 3 is only compatible with Smartphones with Android operating systems at this stage of testing.

My Journey 3 features four key elements of self-management, an approach with demonstrated efficacy in improving social and clinical outcomes for people with psychosis.[12] Screenshots of the key components are displayed in figure 1. Users can create a relapse prevention plan, where there is the opportunity to identify and list triggers, early warning signs of relapse and personalised coping strategies to refer to as required and to create a plan to follow if experiencing a crisis. Via the 'My Recovery Plan' section, users are able to set recovery goals, list actions they can do to encourage well-being and set reminders on their Smartphone to encourage engagement in these activities. Users can also use a tracker to monitor and rate their symptoms and early warning signs over time. In the Symptom Tracker, users are presented with 17 different symptoms and behaviours and are asked to respond via a "Yes/No" format as to whether they have recently experienced these. Users who respond with a

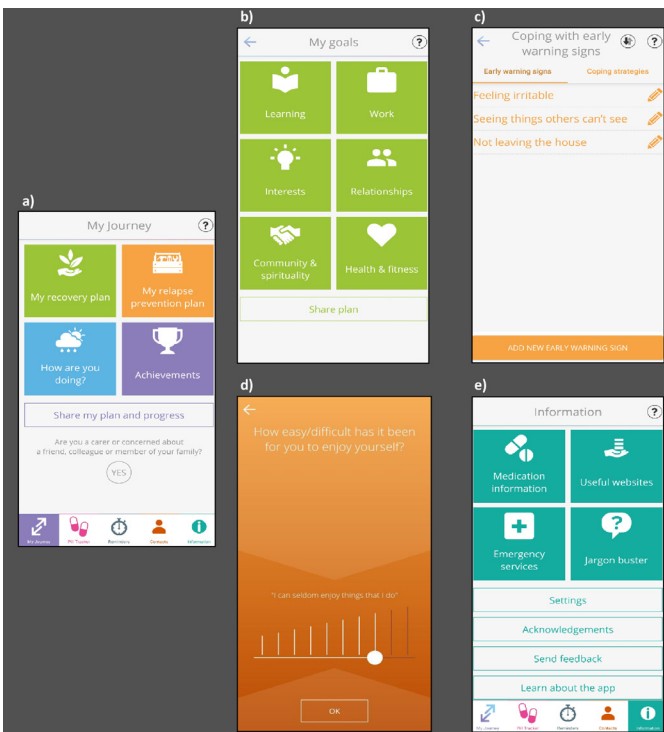

**Figure 1** Screenshots of the My Journey 3 app. (A) The homescreen, (B) the 'My goals' section of the recovery plan, (C) the 'Coping with early warning signs' section of the relapse prevention plan, (D) an example item from the Symptom Tracker, (E) the Information section.

"Yes" are then presented with a 10-point scale (4-point scale for the early warning sign tracker) to rate the severity or frequency of the associated symptoms, with advice on how to manage these symptoms displayed. Psychoeducation on mental health, medication and mental health services is provided in an 'Information' section. To encourage

adherence with medication, users are encouraged to log and track their medication in the 'Pill Tracker' section. Users are able to set daily alerts to remind them to log whether they have taken their medication. My Journey 3 also features weekly discrete notifications to encourage engagement with the app, which can be disabled at the users' preference. The key components of My Journey 3 are summarised in table 1, with further details available in the protocol paper.[32]

Prior to the feasibility trial reported in this paper, My Journey 3 was tested by EIP service users in laboratory-based usability tests and in a 1-month field study. The final content of My Journey 3 was then refined based on feedback from individual interviews with the participating EIP service users and clinicians. No changes were made to the content of My Journey 3 during the feasibility RCT. A major technical update to My Journey 3 was carried out in January 2018 to fix compatibility issues with older versions of Android operating systems. This did not require any changes to the trial design.

### Delivery

Following assignment to the treatment group, participants engaged in individual training sessions with a trial researcher and a supporting EIP service clinician. Training sessions were intended to take place within 6 weeks of the participants' initial baseline assessment, and lasted for approximately 2 hours. During these sessions, the researcher downloaded My Journey 3 onto the participants' Smartphone and gave a demonstration of the app and its main functions. Participants were then encouraged to input appropriate information to specific sections of My Journey 3 with the help of the supporting EIP service clinician in attendance. Following this session, it was hoped that all participants had initial personal

| Table 1 | Key sections of the My Journey 3 Smartphone app | |
|---|---|---|
| **Section** | **Features** | **Purpose** |
| My recovery plan | Things I can do to keep well<br>My goals | To encourage users to have regular routines, track activities, set reminders and plan how to achieve long-term goals |
| My relapse prevention plan | Coping with triggers<br>Coping with early warning signs<br>Coping with a crisis<br>Crisis contacts | To help users identify, monitor and cope with triggers and early warning signs<br>To help users create a 'relapse plan' to follow in times of crisis |
| How are you doing? | My mood<br>My early warning signs<br>My tracker | For users to monitor symptoms, behaviours and early warning signs and track these experiences over time |
| Pill tracker | | To log whether users have taken their medication each day |
| Information | Medication information<br>Useful websites<br>Emergency services<br>Jargon buster | To provide users with useful information and external links on medication and mental health<br>To identify local emergency services in a time of crisis<br>To provide a glossary of terms that are commonly used in mental health care |

recovery plans, relapse prevention plans and crisis plans stored on My Journey 3.

Participants had access to My Journey 3 on their own Smartphone from the training session until the 12-month time point. Researchers recommended that participants used My Journey 3 at least once a week, but participants had a free choice in how and when they used My Journey 3. Participants did not receive any financial incentives to use My Journey 3, and were free to withdraw from using the app or decline the installation of it on to their Smartphone. At the training session, participants were informed by the researcher that My Journey 3 would be not suitable for seeking urgent medical care while in crisis, and that it is not a substitute for human support.

To encourage user engagement with My Journey 3 during the trial, supporting EIP service clinicians were asked to provide regular support and encouragement to service users who had access to My Journey 3. Clinicians were asked to discuss recovery goals and relapse prevention plans in routine appointments with participants, and assist with entering these into the appropriate My Journey 3 sections. Clinicians had an existing understanding of self-management approaches from their clinical training and practice, and would be able to provide appropriate advice with the intervention components of My Journey 3, but they received no formal training on how to implement My Journey 3 into their clinical work. Clinicians' understanding of operating My Journey 3 was from the training sessions only. Clinician support for My Journey 3 as part of the trial was not manualised or incentivised.

Participants were encouraged to contact the trial researcher in the case of technical problems with My Journey 3. The researcher contacted participants a week after the training session to check that My Journey 3 had been functioning without issues and invited any questions about the app. No further prompts were instigated by the researcher during the trial.

## Treatment as usual

All participants received TAU regardless of group allocation. TAU for a person under the care of EIP services typically involves regular meetings with a care coordinator, access to a psychiatrist, psychiatric medication and a range of psychological interventions. EIP services are encouraged to deliver self-management programmes that include advice on symptom management, crisis planning and relapse prevention, generally delivered with paper-and-pen tools if at all.[34] None of the participating EIP services offered digital interventions or Smartphone apps as part of routine care during the study period, and structured self-management support, including the relapse prevention work recommended in EIP contexts, was inconsistently implemented.

## Patient and participant involvement

The development of My Journey 3 has been guided by the input of people with lived experience of psychosis. Initial development of the design and content involved a collaboration between researchers, experts in digital health and service users. Service users provided further input into the design and functionality of My Journey 3 by providing feedback after taking part in laboratory-based tests and a field study.

## Outcomes

Participant data were collected from numerous sources including participant assessments, patient records and anonymous My Journey 3 usage reports. There were no pre-specified criteria for assessing trial feasibility and intervention acceptability.

### Questionnaire measures

Proposed outcome measures for a future trial were assessed at structured face-to-face assessments with a trained researcher at three time points: baseline, 4 months post baseline and 12 months post baseline. At all meetings, participants completed self-report questionnaires that have been previously used with people with first-episode psychosis. Participants were given £20 as a thank you for completing the assessment at each time point.

At each assessment, we collected sociodemographic data including age, gender, ethnicity, accommodation and living situation, employment status, educational attainment, Smartphone use and use of other mental health apps. The following self-report measures were also collected: social outcomes (Social Outcomes Index (SIX),[38] score 0–6: higher score=better social outcomes), self-efficacy (Mental Health Confidence Scale (MHCS),[39] score 16–96: higher score=greater empowerment), self-rated recovery (Questionnaire about the Process of Recovery (QPR),[40] intrapersonal score 0–68, interpersonal score 0–20: higher score=greater recovery), mental well-being (Warwick-Edinburgh Mental Well-Being Scale (WEMWBS),[41] score 14–70: higher score=greater well-being) and quality of life and satisfaction with treatment (DIALOG scale,[42] score 1–7: higher score=greater quality of life/satisfaction with treatment).

Clinical structured interviews were also conducted with each participant by the researcher, to assess psychopathology, using the Positive and Negative Syndrome Scale (PANSS).[43] Higher PANSS scores are indicative of greater severity of each symptom domain.

Participants' engagement with EIP services were measured using the Service Engagement Scale (SES),[44] completed by EIP service clinicians known to each participant, typically care co-coordinators. Clinicians completed the SES at baseline and 12 months later, regardless of whether participants attended the 12-month assessment. Higher SES scores are indicative of poorer user engagement with EIP services.

### Patient records

Clinical data were extracted from patient records at baseline and at the 12-month time point. Clinical measures included most recent diagnosis and use of EIP services,

other community mental health teams and acute mental health services in the previous 12 months.

The proposed primary outcome for a future RCT (relapse of psychosis) was operationalised as an admission to an acute mental health service (inpatient psychiatric ward, crisis house, crisis resolution team or acute day care service) during the 12-month trial period as indicated in patient records. This definition of relapse has been used previously in a recent trial of a self-management intervention.[45]

### My Journey 3 use

To assess acceptability of the intervention and user engagement, My Journey 3 usage data were collected for all participants in the treatment group from the training session until the 12-month time point. Whenever users had Wi-Fi internet access on their Smartphone, My Journey 3 automatically uploaded encrypted usage data to a secure server. Data collected included a record of each time the user opened My Journey 3, whether this was in response to a prompt and which components they used. To ensure confidentiality, personal or identifiable data such as text or responses to each sections were not collected.

### Acceptability

Feedback was obtained through semi-structured interviews as part of a nested qualitative study. Individual interviews were conducted at the 4-month time point with both service user participants that received My Journey 3 and supporting clinical staff.

### Analysis

Participant demographic and clinical characteristics, My Journey 3 usage, and rates of participant recruitment and retention were summarised using descriptive statistics. As this was a feasibility RCT, it was not powered to assess the effectiveness of the intervention. Statistical analyses of participant outcome measures were conducted to pilot the methods of analysis for a fully powered effectiveness trial. Logistic regression was used to explore the impact of the My Journey 3 intervention on relapse. Linear regression was used to examine the potential effect of the intervention on continuous outcome measures at 4 months and 12 months separately. We report the effect estimates and corresponding 95% CIs only for unadjusted analyses and for analyses adjusting for the baseline measure of the outcome in question. All analyses were performed using STATA V.14 after completion of the final participant assessment. No interim analyses were conducted.

Qualitative data were coded to themes based on the Acceptability of Healthcare Interventions framework.[46] Results of the nested qualitative study exploring the acceptability of My Journey 3 and drivers of engagement and non-adherence will be reported in full elsewhere. Here, we provide a short summary of findings.

## RESULTS

### Feasibility of trial design

Participant flow is detailed in the CONSORT diagram (figure 2). A total of 40 participants was recruited and randomised (20 to My Journey 3, 20 to TAU) over a 7-month period from March 2017 to September 2017. Participants were recruited until the required number of 40 was obtained: we do not therefore have a full assessment of the proportion of the teams' caseload who could have been recruited to a full trial, nor do we know the proportion of approached EIP services users that did not meet eligibility criteria or declined involvement in the trial.

Among those recruited to the trial, attrition rates were generally low: 83% (33/40) and 75% (30/40) of participants successfully attended and completed follow-ups at 4 months and 12 months, respectively. At both time points, the follow-up rate was lower in the control group (4 months: 65% compared with 100%, 12 months: 70% compared with 80%). Patient record data were available for all participants at baseline and for 95% of the sample (38/40) at the 12-month time point. Completion rates of the SES by clinicians were higher at baseline (90%) than at the 12-month time point (67.5%). Follow-up assessments were conducted from July 2017 to October 2018.

All participants in the treatment group attended a training session with a researcher and had access to My Journey 3 during the trial. Issues with Smartphone compatibility initially prevented three participants from downloading My Journey 3. Following an update to the system, two of the participants were able to install and access My Journey 3 on their own Smartphones. Two participants were provided with Smartphones with My Journey 3 pre-installed (the app was still incompatible on one participant's Smartphone despite the update; another participant no longer owned an Android Smartphone after entering the trial). The median length of time from trial enrolment to having access to My Journey 3 was 14 weeks (IQR 11 to 17), longer than the planned time of 6 weeks. Participants had access to My Journey 3 for a median of 38.1 weeks (IQR 34.8 to 40.7). There were no reported privacy breaches.

My Journey 3 usage data were collected for all participants following the training session, with 500 different data entries available for analysis. Within the 500 data entries, 27 (5.4%) were corrupt and were subsequently removed from the analysis. The unusable data can grouped into two types. The first, duplicates of previous data entries that were subsequently removed. The second, entries where the times were implausible (eg, the end time of using My Journey 3 was recorded as occurring before the start time). In addition, a further issue caused errors with accurately recording My Journey 3 usage data of 'My Recovery Plan' and 'My Relapse Plan' sections. As a result, we were unable to accurately conclude how often participants used these sections.

One participant randomised to the control group was wrongly given access to My Journey 3. For the purpose

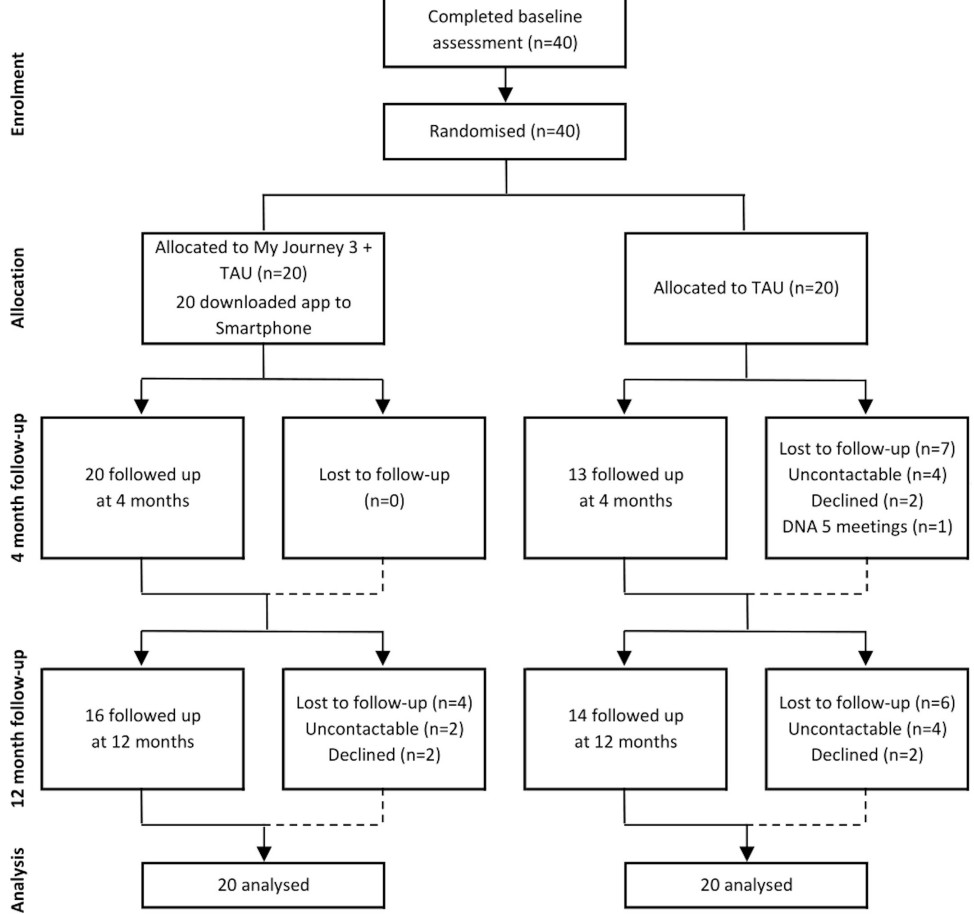

**Figure 2** CONSORT diagram of the ARIES feasibility trial. Note: DNA, did not attend.

of the statistical analysis, they are classed as a control participant.

## Sample characteristics

A summary of demographic and clinical characteristics of the sample is displayed in table 2. The sample was predominantly male (n=28, 70%). The mean age of the sample was 29.7 years (SD 9.78) and similar to that of UK cohorts of EIP service users at first presentation.[47 48] Six participants were over the age of 35, with these participants spread evenly across the two groups. Most participants had a diagnosis of a schizophrenia, schizotypal or delusional disorder (ICD code F20–F29) and were not in paid employment. A quarter of the sample (n=10, 25%) had completed a university degree. Eight (20%) participants had previously used a mental health app.

## My Journey 3 use

The level of My Journey 3 use was highly skewed. The median number of times My Journey 3 was used per participant during the trial was 16.5 (IQR 8.5 to 23). Participants accessed My Journey 3 on a median of 3.22% (IQR 1.89 to 6.36) of the days it was available to them, equating to My Journey 3 being used on average once every 31 days (IQR 15.7 to 52.9). Participants spent a median of 26.8 min (IQR 18.3 to 57.3) in total using My

Journey 3 over the course of the trial. Eight participants (40%) used My Journey 3 for longer than 30 min in total.

Five participants (25%) were still using My Journey 3 six months after downloading it; however, one participant never used the app after the training session (figure 3). Half of the participants (n=10) stopped using My Journey 3 within the first 3 months after the training session.

The average number of uses by participants for each My Journey 3 component is displayed in table 3. The most frequently accessed section was the "How are you doing?" Symptom Tracker section (median uses 3; IQR 1 to 6); however, data on how frequently users accessed 'My Recovery Plan' and 'My Relapse Plan' are unavailable. The 'Information' section was accessed the fewest times, with 25% (n=5) of participants in the treatment group never using that section following the training session. Just over 7% of My Journey 3 uses were initiated following a prompt from the app.

## My Journey 3 acceptability

Qualitative interviews were conducted with all participants who received My Journey 3 and the majority of clinical staff who supported its delivery. In general, most service user participants found My Journey 3 to be acceptable, and a number of participants reported a clear benefit from using it. Barriers affecting use were identified including

**Table 2** Key demographic and clinical characteristics of the sample at baseline

| | Control (n=20) | My Journey 3 (n=20) |
|---|---|---|
| **Age (years)—mean (SD), (min, max)** | 30 (10.1), (18.8, 64.7) | 29.4 (9.7), (17.6, 52.4) |
| **Gender** | | |
| Female | 7 (35%) | 5 (25%) |
| **Ethnicity** | | |
| White British | 6 (30%) | 8 (40%) |
| Any other white/Mixed white | 2 (10%) | 1 (5%) |
| Black African | 5 (25%) | 3 (15%) |
| Black Caribbean | 1 (5%) | 1 (5%) |
| Black Other | 1 (5%) | 0 |
| Asian Indian | 1 (5%) | 0 |
| Asian Other | 1 (5%) | 2 (10%) |
| Other/Mixed other | 3 (15%) | 3 (15%) |
| **Education** | | |
| Undergraduate degree | 6 (30%) | 4 (20%) |
| Some University but no degree | 3 (15%) | 2 (10%) |
| Higher National Degree or professional qualification | 2 (10%) | 1 (5%) |
| A Levels or equivalent | 3 (15%) | 4 (20%) |
| GCSEs or equivalent | 4 (20%) | 6 (30%) |
| No qualifications | 1 (5%) | 3 (15%) |
| Missing | 1 (5%) | 0 |
| **Employment status** | | |
| Employed—more than 16 hours a week | 4 (20%) | 4 (20%) |
| Employed—less than 16 hours a week | 0 | 2 (10%) |
| Voluntary work | 3 (15%) | 3 (15%) |
| In study or training | 1 (5%) | 1 (5%) |
| Unemployed or exempt due to disability | 8 (40%) | 8 (40%) |
| Missing | 4 (20%) | 2 (10%) |
| **Primary diagnosis (ICD-10 code)** | | |
| F10–F19: Mental and behavioural disorder due to psychoactive substance use | 1 (5%) | 0 |
| F20–F29: Schizophrenia, schizotypal and delusional disorder | 16 (80%) | 13 (65%) |
| F30–F39: Mood disorder | 1 (5%) | 5 (25%) |
| Missing | 2 (10%) | 2 (10%) |
| **Admission to an acute mental health service in previous year** | | |
| Yes | 11 (55%) | 10 (50%) |
| **SIX—mean (SD), (min, max)** | 3.2 (1.5), (0, 6) | 3.6 (1.5), (1, 6) |
| **MHCS—mean (SD), (min, max)** | 59.7 (17.8), (16, 82) | 61.2 (12.6), (38, 78) |
| **QPR—mean (SD), (min, max)** | | |
| Intrapersonal | 45.7 (12), (22, 68) | 42.2 (10.6), (24, 60) |
| Interpersonal | 13.7 (2.7), (9, 19) | 12.9 (3.4), (5, 19) |
| **WEMWBS—mean (SD), (min, max)** | 43.4 (11.6), (25, 69) | 40.3 (10.2), (23, 57) |
| **DIALOG—mean (SD), (min, max)** | | |
| Quality of life | 4.5 (1), (2.8, 6.5) | 4.4 (0.8), (3, 5.7) |
| Treatment satisfaction | 5.4 (0.7), (4.3, 7) | 4.8 (0.7), (3.7, 6) |

Continued

**Table 2** Continued

| | Control (n=20) | My Journey 3 (n=20) |
|---|---|---|
| **PANSS—mean (SD), (min, max)** | | |
| Positive | 10.9 (5), (7, 22) | 11.3 (4.2), (7, 19) |
| Negative | 10.7 (2.5), (7, 19) | 11.8 (4.5), (7, 20) |
| General | 26.6 (6), (17, 39) | 26.2 (8), (16, 46) |
| **SES—mean (SD), (min, max)** | 11.3 (7.9), (0, 26) | 9.6 (7), (0, 23) |

All statistics are reported N (%) unless otherwise specified. Missing data: PANSS scores—one control group participant, SES—three control group participants, one treatment group participant.

a lack of clinician support and concerns around data privacy. A key theme for staff was that they often did not have the time to provide regular support to participants with My Journey 3.

### Participant outcomes

No research-related serious adverse events were recorded. Psychotic and general symptoms (measured by the PANSS) were generally low at all times for both groups, suggesting a stable sample. Summary statistics and estimated effect sizes of participant outcomes are displayed in table 4. Inspection of the effect sizes and confidence intervals suggest that were no obvious differences for any outcome measure between the treatment and control group at either time point.

Of the 38 participants whose patient records data were available, only five experienced a relapse during the trial, as indicated by using an acute mental health service. In the treatment group, 15% of participants (3/20) experienced

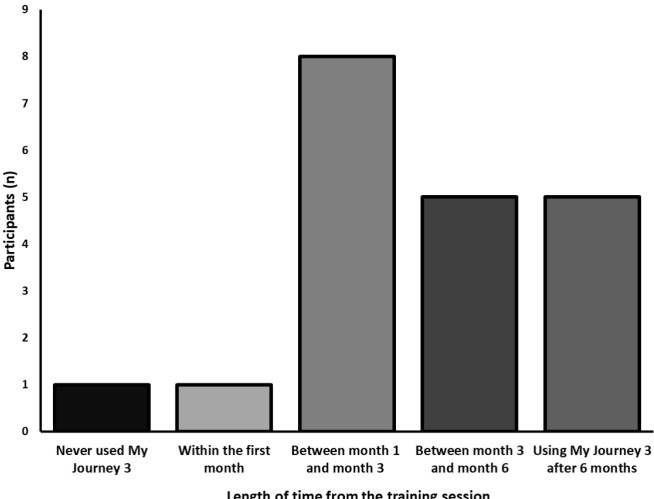

**Figure 3** Bar chart displaying how long after the training session participants disengaged with My Journey 3. For the participants aged over 35, one participant disengaged in the first month (second column), one between 3 and 6 months (fourth column) and the other 35+ participant was still using My Journey 3 six months after the training session (fifth column).

a relapse during the trial period compared with 11% (2/18) in the control group. We found no evidence of a difference in relapse between the two groups (OR 1.41; 95% CI 0.21 to 9.58), but did not have sufficient power for an informative test.

### DISCUSSION

The present study examined the feasibility of conducting an RCT of a supported self-management Smartphone app in EIP services. My Journey 3 aims to facilitate recovery and prevent relapse primarily via the digital delivery of previously developed paper-and-pen self-management tools. The trial indicates that recruitment and retention in an RCT evaluating My Journey 3 is feasible, and that My Journey 3 can be delivered in EIP services. The level of My Journey 3 use was relatively low across the trial period.

Building on from extensive preliminary work with NHS staff and service users, adults with lived experience of psychosis and experts in digital health, we were able to successfully develop a self-management Smartphone app that can be used in EIP services. My Journey 3 appeared to be safe with no related serious adverse events reported. My Journey 3 was successfully delivered to all participants in the treatment group; however, technical problems with the intervention caused significant delays in providing access. Prior to any future evaluations, technical problems with My Journey 3 will need to be identified and fixed to ensure the intervention is implemented as intended.

My Journey 3 use varied considerably between participants, with only a small proportion of participants frequently engaging with the app after obtaining access to it. This raises questions about whether use was at a level where it is likely that useful self-management activities were taking place: certainly not enough time was spent regularly enough for participants to be engaging in detailed monitoring of symptoms and early warning signs, tracking medication and activities and referring to crisis or recovery plans. Despite that, 40% of participants used My Journey 3 for a minimum of 30 minutes which could be an adequate amount of time for users to effectively monitor relapse signs and follow a crisis plan when needed. We have not found evidence on how regularly EIP service users make use of pen-and-paper self-management

**Table 3** Participant use of My Journey 3 and various sections

| | Number of times used per participant | Days used while having access to My Journey 3 (%) | Participants that did not use app or section—n (%) |
|---|---|---|---|
| My Journey 3 | 16.5 (8.5 to 23) | 3.22 (1.89 to 6.36) | 1 (5%) |
| How are you doing? | 3 (1 to 6) | 1.08 (0.4 to 2.12) | 3 (15%) |
| Pill tracker | 2 (1 to 3.5) | 0.73 (0.36 to 1.07) | 3 (15%) |
| Information | 1 (0 to 2.5) | 0.48 (0.18 to 0.7) | 5 (25%) |

All median (IQR), except when stated.

interventions delivered in routine settings, and this was not measured in our trial. Long-term engagement with My Journey 3 appears a challenge, but low levels of app use is a common phenomenon with market research showing that 62% of users stop using Smartphone apps after 10 or fewer uses.[49]

Age has been shown to be an important factor linked to engagement with mental health apps and general Smartphone use,[50] and could partially explain differences in user engagement of My Journey 3. The treatment group, however, featured only a small number of participants from older age groups. We therefore lack informative data regarding app engagement for older participants and we are accordingly unable to explore if engagement and pattern of use of My Journey 3 varied between age groups.

Participant retention for research data collection was high, with 75% of the sample attending the 12-month follow-up assessment, and is comparable with other Smartphone app studies.[51] Completion rates of the SES by EIP service clinicians were much lower at the 12-month follow-up in comparison with baseline, potentially due to staff changes and participants being discharged from services. Recruitment strategies were largely successful; however, data are lacking on overall proportion of caseload recruited, reasons for non-inclusion and the numbers that were assessed for eligibility, thus limiting the conclusions we can make regarding trial feasibility.

The trial was not powered to detect effectiveness, and, as expected with our small number of participants, we found no significant differences between groups on any outcomes, with CIs generally including substantial effects in either direction. Accordingly, we cannot draw any conclusions regarding the potential impact of My Journey 3 as a mental health intervention. The proposed primary outcome for a full-scale trial, relapse as defined by use of an acute mental health service during the trial period, was marked by low event rates. Only five participants (12.5%) experienced a relapse during the 1-year follow-up period, compared with expected levels of 12% to 47%.[52] Consideration should be given to whether relapse, or our measure of relapse, is an appropriate outcome for a future RCT of this intervention. Symptom severity or alternatively patient-valued outcomes of personal recovery that self-management interventions have been shown to benefit may be more suitable primary outcomes in a future large-scale trial.[12]

## Strengths and limitations

My Journey 3 has been developed with extensive stakeholder input, and the intervention has been tested through laboratory testing and a field study prior to the feasibility RCT. In comparison with previous studies,[51] participants had access to the app for a longer period of time. Participants' app use and usage data may be more reflective of real-world use as a result. Participant data were also collected from a wide range of methods including from participant assessments and patient records. The proposed primary outcome for a future RCT (relapse) was measured objectively and data were obtained for 95% of participants.

We recruited until the required number of participants was obtained rather than screening caseloads objectively: as a result, we are not aware of the proportion eligible who were recruited, reasons for non-eligibility and how many EIP service users declined to take part and why. This limits our understanding of how feasible conducting a large-scale trial of this intervention would be. In addition, there were problems with the usage data, which impacts the reliability of our conclusions regarding how often participants engaged with My Journey 3.

The trial did not feature an active digital placebo for the control group, meaning that non-specifics of Smartphone use could not be controlled for. Furthermore, data were not collected during the study period from either group regarding frequency of completing recovery work such as relapse prevention plans, recovery plans or crisis plans either in paper-and-pen or digital format, limiting our understanding of whether access to My Journey 3 facilitated increased access to self-management activities.

Although clinicians were encouraged to support participants with My Journey 3, support was not manualised and clinicians did not have personal access to the app or associated data, potentially limiting the level and quality of the support offered and therefore user engagement. Future developments of My Journey 3 should focus on effective implementation and delivery within healthcare settings, and there should be considerations on how to facilitate secure data-sharing between My Journey 3 and healthcare records or other secure web-based platforms dependent on user consent, which is likely to increase clinician engagement with the app and its utility.[53]

We did also not define pre-specified criteria for assessing the feasibility of a RCT and the acceptability of My Journey 3. Instead, we will consider all findings from

**Table 4** Summary statistics and unadjusted and adjusted treatment effects

| 4-month scores | Control (n=13) Mean (SD) | My Journey 3 (n=20) Mean (SD) | Unadjusted analysis Estimated difference | 95% CI | Analysis adjusted for baseline score Estimated difference | 95% CI |
|---|---|---|---|---|---|---|
| **SIX** (Social Outcomes) | 3.3 (1.9) | 3.6 (1.3) | 0.29 | −0.84 to 1.43 | 0.16 | −0.6 to 0.92 |
| **MHCS** (Mental Health Confidence) | 66.4 (12.7) | 63 (15.8) | −3.43 | −14.1 to 7.25 | −4.81 | −14.88 to 5.25 |
| **QPR** (Recovery) | | | | | | |
| Intrapersonal | 47.8 (10.6) | 43.2 (12.2) | −4.57 | −13 to 3.87 | −2.01 | −8.43 to 4.49 |
| Interpersonal | 13.9 (2.4) | 13.2 (2.3) | −0.72 | −2.39 to 0.95 | −0.42 | −1.97 to 1.13 |
| **MHCS** (Mental Health Confidence) | 46.1 (9.9) | 44 (11.3) | −2.08 | −9.9 to 5.74 | −0.19 | −7.28 to 6.9 |
| **DIALOG** | | | | | | |
| Quality of life | 4.4 (1.2) | 4.5 (0.6) | 0.07 | −0.58 to 0.71 | 0.18 | −0.38 to 0.74 |
| Treatment satisfaction | 5.4 (0.7) | 5 (0.5) | −0.38 | −0.83 to 0.06 | −0.17 | −0.6 to 0.25 |
| **PANSS** (Symptom Severity) | | | | | | |
| **Positive** | 9.3 (2.9) | 11.4 (5.1) | 2.09 | −1.24 to 5.4 | 1.9 | −0.49 to 4.3 |
| **Negative** | 10 (2.3) | 11.1 (3.9) | 1.05 | −1.51 to 3.62 | 0.54 | −1.6 to 2.67 |
| **General** | 23 (4) | 24 (6.7) | 1.21 | −3.19 to 5.61 | 1.35 | −2.68 to 5.37 |
| 12-month scores | Control (n=14) Mean (SD) | My Journey 3 (n=16) Mean (SD) | Unadjusted analysis Estimated difference | 95% CI | Analysis adjusted for baseline score Estimated difference | 95% CI |
| **SIX** (Social Outcomes) | 3.2 (1.9) | 3.5 (1.5) | 0.29 | −0.97 to 1.54 | 0.29 | −0.73 to 1.3 |
| **MHCS** (Mental Health Confidence) | 66.2 (14.1) | 71.1 (12.1) | 4.81 | −5 to 14.62 | 3.03 | −6.04 to 12.1 |
| **QPR** (Recovery) | | | | | | |
| Intrapersonal | 47.3 (11.5) | 49.5 (11.1) | 2.2 | −6.25 to 10.7 | 3.21 | −4.12 to 10.5 |
| Interpersonal | 13.6 (3.4) | 15.1 (3.3) | 1.44 | −1.09 to 3.96 | 1.62 | −0.89 to 4.12 |
| **MHCS** (Mental Health Confidence) | 45.6 (11.3) | 49.3 (9.7) | 3.61 | −4.24 to 11.46 | 5.03 | −1.67 to 11.7 |
| **DIALOG** | | | | | | |
| Quality of life | 4.7 (0.9) | 5 (0.7) | 0.28 | −0.31 to 0.87 | 0.24 | −0.33 to 0.81 |
| Treatment satisfaction | 5.3 (1) | 5.2 (1.2) | −0.12 | −0.93 to 0.69 | 0.31 | −0.42 to 1.04 |
| **PANSS** (Symptom Severity) | | | | | | |
| Positive | 9.5 (2.1) | 10.2 (2.1) | 0.69 | −0.98 to 2.36 | 0.88 | −0.62 to 2.38 |
| Negative | 10.2 (2.2) | 10.9 (3.3) | 0.77 | −1.51 to 3.05 | 0.14 | −1.56 to 1.84 |
| General | 23.5 (5.4) | 22.1 (3.5) | −1.38 | −4.82 to 2.07 | −1 | −4.57 to 2.55 |
| **SES** (Engagement with Services) | 10 (6.2) | 9.5 (8) | −0.4 | −6.08 to 5.28 | 3.11 | −1.57 to 7.79 |

| Table 4 | Continued | | | | | |
|---|---|---|---|---|---|---|
| | Control (n=14) | My Journey 3 (n=16) | Unadjusted analysis | | Analysis adjusted for baseline score | |
| 12-month scores | Mean (SD) | Mean (SD) | Estimated difference | 95% CI | Estimated difference | 95% CI |

Estimated differences and associated 95% Confidence Intervals from linear regression models with the control group as reference. Missing data: 4-month PANSS scores – one control group participant, one treatment group participant. 12-month PANSS scores – two control group participants. Note: 12-month SES data available for 13 control group participants, and 14 treatment group participants.

the trial, app usage data and feedback from qualitative interviews yet to be reported in determining whether My Journey 3 will be evaluated in a full-scale trial. This allows all data from the RCT to be thoroughly considered, but may be a less objective approach in determining feasibility than using pre-defined criteria. Although the trial was not designed to assess intervention effectiveness, participants and trial researchers were not blinded to group allocation, and as such could have led to an inflation of any observed effects.

Finally, the sample consisted of Android Smartphone users who were generally stable and in an appropriate stage of recovery to consider using a self-management Smartphone app. Participants may therefore not be representative of all EIP service users. Furthermore, contact with a researcher within a trial context could have led to increased intervention engagement that would not occur in a real-world clinical environment.

## CONCLUSIONS

We developed and delivered a self-management Smartphone app for first-episode psychosis in a trial context. Participants were successfully recruited, most engaged at least to some extent with the intervention, and they had high follow-up rates over the 1-year trial period. Based on the data presented, the trial methods appear feasible. My Journey 3 was shown to be safe, but the level of use was lower than anticipated thus potentially limiting its utility.

If My Journey 3 is to be further tested in a research setting, attention needs to be given to engagement, a challenge associated with many digital tools in mental health.[54] Further usability testing in laboratory and field settings may be a means to improving engagement. Other potential strategies include making more efforts to engage clinicians as well as service users with My Journey 3 by giving them access to the tool and to aspects of the planning and monitoring that service users conduct through it. The app could also potentially be offered as part of a blended approach to self-management, with pen-and-paper tools also used and as a whole service strategy for implementation of self-management. Refinements required before participating to a full trial including participant and assessor blinding and manualised clinician support should be considered prior to conducting a future RCT.

**Acknowledgements** The ARIES research team are grateful to their software collaborators MyOxygen for their technical development and hosting of My Journey 3 and to Ali Mousa for his valuable contribution to the development of the original My Journey app. We are grateful to Max Birchwood for his permission to incorporate 'Back in the Saddle' into My Journey 3. We are grateful to Rachel Perkins for her permission to adapt the Personal Recovery Plan resource and incorporate in to My Journey 3.

**Contributors** SJ is the chief investigator, based at University College London, DO the co-chief investigator and TS the project manager. The trial design was developed by SJ, DO, BL-E and PO. SA, HR, PO and ME have led on the development of the intervention. TS conducted the statistical analysis, with advice from RJ. TS wrote the draft of the paper, which was revised and approved by all authors. All authors approved the final manuscript.

**Funding** The research is funded by the National Institute for Health Research (NIHR) Collaboration for Leadership in Applied Health Research and Care North Thames at Barts Health NHS Trust (NIHR CLAHRC North Thames). SJ, DO and BL-E are supported by the NIHR Mental Health Research Policy Unit, the NIHR Collaboration for Leadership in Applied Health Research and Care (CLAHRC) North Thames and the UCLH Biomedical Research Centre.

**Disclaimer** The views expressed in this article are those of the authors and not necessarily those of the NHS, the NIHR or the Department of Health and Social Care.

**Competing interests** None declared.

**Patient consent for publication** Not required.

**Ethics approval** National Research Ethics Service Committee London—Brent (Research Ethics Committee reference: 15/LO/1453).

**Provenance and peer review** Not commissioned; externally peer reviewed.

**Data availability statement** No data are available. The datasets generated during and/or analysed during the current study will be available 2 years after the trial end.

**ORCID iDs**
Thomas Steare http://orcid.org/0000-0002-3881-2018
Sonia Johnson http://orcid.org/0000-0002-2219-1384

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
