## [Reviewer comments · BMJ Open]

ARTICLE DETAILS

TITLE (PROVISIONAL)	Smartphone-delivered self-management for first-episode psychosis: the ARIES feasibility randomised controlled trial
AUTHORS	Steare, Thomas; O'Hanlon, Puffin; Eskinazi, Michelle; Osborn, David; Lloyd-Evans, Brynmor; Jones, Rebecca; Rostill, Helen; Amani, Sarah; Johnson, Sonia

VERSION 1 – REVIEW

REVIEWER	Laura M. Tully, Ph.D. University of California, Davis Dr. Tully is co-founder and ownership share holder of Safari Health Inc.
REVIEW RETURNED	16-Nov-2019

GENERAL COMMENTS	Summary: The authors report results of a feasibility trial (two arms, unblinded) of a self-management app for use in EIP services in the UK. Results indicate good feasibility but low usage. The manuscript is well written and clearly describes the study and methods used, with clear and open discussion of the study limitations. The study adds value to a growing literature on digital health tools for EIP care and warrants publication. I have only a few suggestions for edits, detailed below. I look forward to seeing additional work from this group on this application. Introduction: 1. Recommend clarification around the distinction between apps for “self-management” and “within EIP services”. “Self-management” implies using the app independent (or in the absence of) outpatient services (e.g., FOCUS from Ben-Zeev’s group, Danielle Schlosser’s work at UCSF) whereas “within EIP services” implies using the app as part of clinical care in outpatient services (e.g., Niendam & Tully et al. 2018 work). It’s not clear if the author’s are proposing a wholly self-management app or an approach wholly integrated with current services or some blend of the two? Recommend clarifying this in the intro. 2. Pg. 6. Authors state “only one trial of self-management app for EIP services has published its results”. Recommend editing to state that although several self-management apps have been tested for feasibility and acceptability either as part of or independent to EIP services (e.g., See Torous, Woodyatt & Tully 2019 for a review) , only one study has published trial results comparing the app to passive control app. This will maintain the point the authors are making re: the lack of trial data whilst still acknowledging that a fair amount of work is being done in this area. Methods: 1. Was the EIP service clinician present for the training session?
--

	(i.e., how were clinicians trained in using and implementing the app as part of clinical care?). Recommend adding some detail on this. 2. Did the study track whether personal recovery plans, relapse prevention plans, and crisis plans were in place for the control arm as well (albeit not in an app)? Similarly, did the study track how often clinicians discussed these issues with clients in each arm? Recommend clarifying this and acknowledging this as a limitation if not tracked in both (or either) arms. 3. P16. clarify what “most recent care cluster” means. 4. Did My Journey 3 send notifications to users to remind the to use it? (“prompts” are mentioned in the usage data export but I don’t see them mentioned in the description of the app). Recommend adding detail regarding frequency and nature of prompts/notifications. Results: 1. Recommend adding minimum and maximum values (where appropriate) to table 2. 2. App usage: what proportion of usage was in response to a prompt / during a clinic appointment vs. initiated spontaneously by the user? If average usage happened once every 31 days, that could coincide with monthly clinic visits, suggesting that app usage is reliant on scaffolding from clinicians/support staff. (relatedly, is the information on frequency of clinic visits over the course of the study? this would help clarify this issue). Discussion: 1. The authors indicate that relapse - as defined by a hospitalization - may not be an appropriate outcome for large scale trials given its low frequency in this sample. I agree - we have seen similarly low frequencies in our data, and have also looked at frequency of the return of psychotic level symptoms (e.g. 100% conviction in a delusional belief) in the absence of hospitalization, as well as exploring other indicators of symptom exacerbation and functional decline (e.g., suicidal ideation or behavior that doesn’t warrant hospitalization, family members intervening to prevent hospitalization). Do the authors have suggestions for alternate outcomes? Recommend including a short discussion if this issue. 2. Regarding low usage, the authors note that clinicians did not have access to the app. Adding in a web-based component that is accessible by clinicians might assist clinicians in supporting app use in patients. Some discussion of how other platforms/studies have worked to increase and maintain usage would be good here.
--	---

REVIEWER	LUCIA BONET MORA Department of Psychiatry, Faculty of Medicine, University of Valencia, Spain
REVIEW RETURNED	21-Nov-2019

GENERAL COMMENTS	This paper highlights the great potential of new technologies in the field of mental health. There is an exhaustive work behind the development of My Journey 3 app and authors provide enough and accurate information to validate its use. However, there are some minor suggestions I would like to make. I feel some extra information regarding the functioning of the app would make the intervention easier to understand. I do not quite get how do patients create the plans (Lines 19-25: recovery plan, relapse prevention, etc.). Moreover, there are some other issues that are not clear from my point of view: How are the assessing items presented and how do patients provide their health information
--

	(yes/no questions, short answers, multiple choice questions, etc.)? I highly recommend including some screenshots of the app or a brief schema of the information gathered by the app.
REVIEWER	Brittany Lapin Cleveland Clinic USA
REVIEW RETURNED	10-Dec-2019
GENERAL COMMENTS	This is a RCT to assess feasibility and acceptability. The objectives are clear, statistical methods are appropriate, and results are well-presented. Testable hypotheses are required though. It is unclear how feasibility and acceptability will be demonstrated. Thresholds should be stated by which the study results will show the intervention is feasible/acceptable (ie use by x% over x months will demonstrate acceptability, etc). Also, please clarify the Table 4 estimated differences in a footnote. Are these the estimates from regression models for the My Journey 3 group with the control group as reference?

VERSION 1 – AUTHOR RESPONSE

Reviewer(s)' Comments to Author:

Reviewer: 1

Reviewer Name: Laura M. Tully, Ph.D.

Institution and Country: University of California, Davis Please state any competing interests or state 'None declared': Dr. Tully is co-founder and ownership share holder of Safari Health Inc.

Please leave your comments for the authors below

Summary: The authors report results of a feasibility trial (two arms, unblinded) of a self-management app for use in EIP services in the UK. Results indicate good feasibility but low usage. The manuscripts is well written and clearly describes the study and methods used, with clear and open discussion of the study limitations. The study adds value to a growing literature on digital health tools for EIP care and warrants publication. I have only a few suggestions for edits, detailed below. I look forward to seeing additional work from this group on this application.

Introduction:

1. Recommend clarification around the distinction between apps for “self-management” and “within EIP services”. “Self-management” implies using the app independent (or in the absence of) outpatient services (e.g., FOCUS from Ben-Zeev’s group, Danielle Schlosser’s work at UCSF) whereas “within EIP services” implies using the app as part of clinical care in outpatient services (e.g., Niendam & Tully et al. 2018 work). It’s not clear if the author’s are proposing a wholly self-management app or an approach wholly integrated with current services or some blend of the two? Recommend clarifying this in the intro.

Our response: Thank you for this helpful suggestion. Edits have been made to the introduction to clarify that My Journey 3 is primarily intended to be embedded into clinical care but that it is also suitable for independent use, especially once support has been given in setting the app up (page 7).

2. Pg. 6. Authors state “only one trial of self-management app for EIP services has published its results”. Recommend editing to state that although several self-management apps have been tested for feasibility and acceptability either as part of or independent to EIP services (e.g., See Torous, Woodyatt & Tully 2019 for a review), only one study has published trial results comparing the app to passive control app. This will maintain the point the authors are making re: the lack of trial data whilst still acknowledging that a fair amount of work is being done in this area.

Our response: Thank you for this suggestion. We have added a statement to give greater emphasis on the existing work on the acceptability and feasibility of Smartphone apps for psychosis (page 6).

Methods:

1. *Was the EIP service clinician present for the training session? (i.e., how were clinicians trained in using and implementing the app as part of clinical care?). Recommend adding some detail on this.*

Our response: We have added greater detail on the training session, explaining that the clinicians were present at the training session but that they did not receive further training on how to implement My Journey 3 into clinical care (page 14). Thank you for the suggestion.

2. *Did the study track whether personal recovery plans, relapse prevention plans, and crisis plans were in place for the control arm as well (albeit not in an app)? Similarly, did the study track how often clinicians discussed these issues with clients in each arm? Recommend clarifying this and acknowledging this as a limitation if not tracked in both (or either) arms.*

Our response: Thank you for this suggestion. We have added to the discussion how the lack of data on the completion of personal recovery plans, relapse prevention plans, and crisis plans by participants is a limitation of the study (page 26).

3. *P16. clarify what “most recent care cluster” means.*

Our response: We have removed the text referring to “most recent care cluster” from the manuscript, as it was not referred to elsewhere and is redundant.

4. *Did My Journey 3 send notifications to users to remind them to use it? (“prompts” are mentioned in the usage data export but I don’t see them mentioned in the description of the app). Recommend adding detail regarding frequency and nature of prompts/notifications.*

Our response: Thank you for the suggestion. We have added greater detail to the methods section to provide more information regarding the use of prompts by My Journey 3 (page 12).

Results:

1. *Recommend adding minimum and maximum values (where appropriate) to table 2.*

Our response: Thank you for the helpful suggestion. We have added minimum and maximum values for appropriate variables in table 2 (page 20).

2. *App usage: what proportion of usage was in response to a prompt / during a clinic appointment vs. initiated spontaneously by the user? If average usage happened once every 31 days, that could coincide with monthly clinic visits, suggesting that app usage is reliant on scaffolding from clinicians/support staff. (relatedly, is the information on frequency of clinic visits over the course of the study? this would help clarify this issue).*

Our response: We are in the process of writing a separate paper reporting findings from qualitative interviews with participants and supporting staff. This will explore barriers and facilitators of engagement, including the level of support from clinicians. This paper will examine issues of clinician support in detail, and we plan to make use of this helpful suggestion for this second paper.

Discussion:

1. *The authors indicate that relapse - as defined by a hospitalization - may not be an appropriate outcome for large scale trials given its low frequency in this sample. I agree - we have seen similarly low frequencies in our data, and have also looked at frequency of the return of psychotic level symptoms (e.g. 100% conviction in a delusional belief) in the absence of hospitalization, as well as exploring other indicators of symptom exacerbation and functional decline (e.g., suicidal ideation or*

behavior that doesn't warrant hospitalization, family members intervening to prevent hospitalization). Do the authors have suggestions for alternate outcomes? Recommend including a short discussion if this issue.

Our response: Thank you for your suggestion. We agree that this is an important point, and we have accordingly added a discussion point around potential alternative outcome measures for a full scale trial including symptom severity and personal recovery (page 26).

2. Regarding low usage, the authors note that clinicians did not have access to the app. Adding in a web-based component that is accessible by clinicians might assist clinicians in supporting app use in patients. Some discussion of how other platforms/studies have worked to increase and maintain usage would be good here.

Our response: We agree that clinician engagement and involvement is an important issue. We have added greater detail in the discussion section on the importance of data-sharing between clinicians and digital tools, and the potential for greater data-sharing in My Journey 3 (page 27).

Reviewer: 2

Reviewer Name: LUCIA BONET MORA

Institution and Country: Department of Psychiatry, Faculty of Medicine, University of Valencia, Spain

Please state any competing interests or state 'None declared': None declared

Please leave your comments for the authors below This paper highlights the great potential of new technologies in the field of mental health. There is an exhaustive work behind the development of My Journey 3 app and authors provide enough and accurate information to validate its use. However, there are some minor suggestions I would like to make.

I feel some extra information regarding the functioning of the app would make the intervention easier to understand. I do not quite get how do patients create the plans (Lines 19-25: recovery plan, relapse prevention, etc.). Moreover, there are some other issues that are not clear from my point of view: How are the assessing items presented and how do patients provide their health information (yes/no questions, short answers, multiple choice questions, etc.)? I highly recommend including some screenshots of the app or a brief schema of the information gathered by the app.

Our response: Thank you for your helpful comments. We have added greater detail to the methods section to explain how users create recovery and relapse prevention plans on My Journey 3. We have also provided greater detail on how the user navigates the app and enters their health information (page 11 & 12). We have added a figure to visually display My Journey 3 and its key components (figure 1).

Reviewer: 3

Reviewer Name: Brittany Lapin

Institution and Country: Cleveland Clinic, USA Please state any competing interests or state 'None declared': None declared

Please leave your comments for the authors below This is a RCT to assess feasibility and acceptability. The objectives are clear, statistical methods are appropriate, and results are well-presented. Testable hypotheses are required though. It is unclear how feasibility and acceptability will be demonstrated. Thresholds should be stated by which the study results will show the intervention is feasible/acceptable (ie use by x% over x months will demonstrate acceptability, etc).

Our response: Thank you for your helpful comments. We did not devise criteria for establishing intervention acceptability and trial feasibility. The study features a variety of data types to explore the acceptability of the intervention, the feasibility of trial procedures and to identify potential changes to make prior to a full trial. This includes feedback from qualitative interviews which will be reported elsewhere, app usage data and participant recruitment and retention rates. We felt that these different perspectives could not be all successfully captured by a set of criteria. We have added a sentence to the "Strengths and limitations" section of the discussion to explain our approach and that it is a limitation of our study (page 28).

Also, please clarify the Table 4 estimated differences in a footnote. Are these the estimates from regression models for the My Journey 3 group with the control group as reference?

Our response: Thank you for your suggestion. We have added a sentence to the footnote of Table 4 to explain that the displayed estimates are from the regression models with the control group as reference (page 24).

VERSION 2 – REVIEW

REVIEWER	Laura M Tully University of California, Davis, USA Dr. Tully is is co-founder and ownership shareholder in Safari Health Inc.
REVIEW RETURNED	29-Jan-2020

GENERAL COMMENTS	The edits the authors made per reviewer suggestions have strengthened the paper. I have two remaining suggestions prior to publication: 1. With the addition of the minimum and maximum values to table 2, it appears that there were folks in the study ranging from 17 years old to 65 years old. This is a huge age range, and has ramifications for engagement with the app - it is reasonable to hypothesize that there are age effects in terms of amount of use, type of use, and comfort with using smartphone technology as part of daily life/clinical care. Given that engagement with the app is a core question of the study, I strongly recommend the authors analyze their outcomes taking age into account and adjust tables and figures accordingly (e.g., Figure 3 could be more informative if age is included in some manner. Were older folks less likely to use the app? more likely to disengage sooner? Or was it the opposite?). This has implications for interpreting the results and for implementing smartphone tech in clinical care. Relatedly, I recommend adding some more detail in the methods section about the participants age range - although the authors report inclusion of folks older than 16, there is no detail given regarding the typically age range of folks receiving services in the EIP clinics. Typically (and consistent with average age of onset of psychosis), the majority of folks receiving EIP services are adolescents and young adults (and some clinics, especially outside of the UK, limit care provision by age); the onset of psychosis in later adulthood is less common and it is possible that onset in later adulthood requires different treatment needs. Recommend (1) adding more detail about the clinic inclusion criteria so that readers are more aware of who could be in the trial and (2) addressing the issue of age (consistent with any changes in results when including age) in the discussion. 2. Consistent with the addition of minimum and maximum values to table 2, I recommend adding them to all other relevant tables and stats reported throughout the manuscript.
--

REVIEWER	Brittany Lapin Cleveland Clinic
REVIEW RETURNED	18-Jan-2020

GENERAL COMMENTS	The authors have addressed my concerns. This is a solid paper that
--

	will add to the literature.
--	-----------------------------

VERSION 2 – AUTHOR RESPONSE

Reviewer(s)' Comments to Author:

Reviewer: 3

Reviewer Name: Brittany Lapin

Institution and Country: Cleveland Clinic

Please state any competing interests or state 'None declared': None declared

Please leave your comments for the authors below The authors have addressed my concerns. This is a solid paper that will add to the literature.

Our response: Thank you for reviewing this paper, and for your helpful comments on the previous version.

Reviewer: 1

Reviewer Name: Laura M Tully

Institution and Country: University of California, Davis, USA

Please state any competing interests or state 'None declared': Dr. Tully is is co-founder and ownership shareholder in Safari Health Inc.

Please leave your comments for the authors below The edits the authors made per reviewer suggestions have strengthened the paper. I have two remaining suggestions prior to publication:

1. With the addition of the minimum and maximum values to table 2, it appears that there were folks in the study ranging from 17 years old to 65 years old. This is a huge age range, and has ramifications for engagement with the app - it is reasonable to hypothesize that there are age effects in terms of amount of use, type of use, and comfort with using smartphone technology as part of daily life/clinical care. Given that engagement with the app is a core question of the study, I strongly recommend the authors analyze their outcomes taking age into account and adjust tables and figures accordingly (e.g., Figure 3 could be more informative if age is included in some manner. Were older folks less likely to use the app? more likely to disengage sooner? Or was it the opposite?). This has implications for interpreting the results and for implementing smartphone tech in clinical care.

Our response: Thank you for your suggestion and for reviewing our paper. We agree that age may be an interesting variable that could be related to engagement with My Journey 3 and other digital health tools. Within the treatment group of our study, 85% of participants are under the age of 35 with only three participants above this age. As such three participants is too small a sample to be provide any informative data regarding app engagement for older participants. Analyzing the relationship between app engagement and age is not part of our statistical analysis plan and is not an aim of our feasibility study. As such we have not conducted further analyses to investigate the differences in app engagement across ages. We will take this suggestion on board for planning the analyses of a future large scale trial of My Journey 3.

Relatedly, I recommend adding some more detail in the methods section about the participants age range - although the authors report inclusion of folks older than 16, there is no detail given regarding the typically age range of folks receiving services in the EIP clinics. Typically (and consistent with average age of onset of psychosis), the majority of folks receiving EIP services are adolescents and young adults (and some clinics, especially outside of the UK, limit care provision by age); the onset of psychosis in later adulthood is less common and it is possible that onset in later adulthood requires different treatment needs. Recommend (1) adding more detail about the clinic inclusion criteria so that readers are more aware of who could be in the trial and (2) addressing the issue of age (consistent with any changes in results when including age) in the discussion.

Our response:

Thank you for your helpful suggestions. We have made the following three changes following your advice.

First, in the 'Setting' section of the methods, we have stated how in England EIP services accept adults to the age of 65, with the potential to care for adults above this age range in the rare case of this being clinically appropriate (page 8, paragraph 3).

Second, in the 'Sample characteristics' part of the 'Results' section detail has been added around the mean age of the overall sample, and how this closely aligns with previous UK cohort studies of Early Intervention in Psychosis service users (page 20, paragraph 2).

Third, in the 'Discussion' section we have stated how age is an important variable regarding engagement with mental health apps, but due to the small number of participants over the age of 35 in the treatment group we lack meaningful data to explore this in our feasibility trial (page 26, paragraph 2).

2. Consistent with the addition of minimum and maximum values to table 2, I recommend adding them to all other relevant tables and stats reported throughout the manuscript.

Our response:

Thank you for the suggestion. All results regarding engagement with My Journey 3, participant demographics and clinical outcomes in our manuscript present the spread of distribution either through reporting the standard deviation or inter-quartile range. For normally distributed measures, the distribution is completely described by the mean and standard deviation (SD). For this reason, it is conventional to report mean and SD without the addition of the range (or minimum and maximum values). We believe that the inclusion of multiple statistics can make tables and text cumbersome and overly busy, distracting from the key results, while at the same time not adding an appreciable amount of additional information. Where the mean and SD are reported, it is straightforward to calculate a 95% reference range using the formula $\text{mean} \pm 2 \times \text{SD}$. We accordingly have not added further range values to our results.

VERSION 3 – REVIEW

REVIEWER	Laura Tully University of California Davis, USA
REVIEW RETURNED	Ownership in Safari Health Inc.

GENERAL COMMENTS	1. Please identify, either in the main text, figure 3 legend, or figure 3 itself, the use pattern of the 3 individuals over age 35. 2. I strongly recommend inclusion of the acceptability data (e.g., qualitative interviews) in this manuscript (rather than a third paper on the same study). Acceptability is not currently addressed in the manuscript (as raised in prior rounds of review), limiting the interpretation of the feasibility data, including why app usage was so low, and the outcome data.
--

VERSION 3 – AUTHOR RESPONSE

Reviewer(s)' Comments to Author:

Reviewer: 1
Reviewer Name
Laura Tully

Institution and Country
University of California Davis, USA

Please state any competing interests or state 'None declared':
Ownership in Safari Health Inc.

Please leave your comments for the authors below

1. Please identify, either in the main text, figure 3 legend, or figure 3 itself, the use pattern of the 3 individuals over age 35.

Our response: Thank you for your suggestion and for reviewing our paper. We have provided a sentence in the Sample Characteristics section to report that we have a number of participants in the study over the age of 35 (page 20). As helpfully suggested we have added further detail to the legend of Figure 3 to identify when the three participants aged over 35 disengaged with My Journey 3 (page 31).

2. I strongly recommend inclusion of the acceptability data (e.g., qualitative interviews) in this manuscript (rather than a third paper on the same study). Acceptability is not currently addressed in the manuscript (as raised in prior rounds of review), limiting the interpretation of the feasibility data, including why app usage was so low, and the outcome data.

Our response: Thank you for the suggestion. We have added a summary of findings from the qualitative interviews to provide an understanding of the acceptability of My Journey 3 and of key issues affecting usage (page 23). We have added details regarding the nested qualitative study to the methods section (page 18). The findings of the nested qualitative interviews will be shortly submitted for publication, as they cannot be fully incorporated into this manuscript. The separate publication will complement the findings of this paper. We thank you for your helpful comments and suggestions.